# Repetitive Transcranial Magnetic Stimulation (rTMS) in Post-Traumatic Stress Disorder: Study Protocol of a Nationwide Randomized Controlled Clinical Trial of Neuro-Enhanced Psychotherapy “TraumaStim”

**DOI:** 10.3390/brainsci13091274

**Published:** 2023-08-31

**Authors:** Florian Ferreri, Stephane Mouchabac, Vincent Sylvestre, Bruno Millet, Wissam El Hage, Vladimir Adrien, Alexis Bourla

**Affiliations:** 1Department of Psychiatry, Hôpital Saint-Antoine, Sorbonne Université, AP-HP, 75012 Paris, France; florian.ferreri@aphp.fr (F.F.); stephane.mouchabac@aphp.fr (S.M.); vladimir.adrien@aphp.fr (V.A.); 2ICRIN—Psychiatry (Infrastructure of Clinical Research in Neurosciences—Psychiatry), Brain Institute (ICM), Sorbonne Université, INSERM, CNRS, 75013 Paris, France; b.millet@aphp.fr; 3Service de Psychiatrie Adulte de la Pitié-Salpêtrière, Sorbonne Université, AP-HP, 75013 Paris, France; 4Centre Régional de Psychotraumatologie CVL, CHRU de Tours, UMR 1253, iBrain, Université de Tours, INSERM, 37000 Tours, France; wissam.elhage@univ-tours.fr; 5Clariane, Medical Strategy and Innovation Department, 75008 Paris, France; 6NeuroStim Psychiatry Practice, 75005 Paris, France

**Keywords:** post-traumatic stress disorder, transcranial magnetic stimulation, neurostimulation-assisted psychotherapy

## Abstract

The use of high-frequency Transcranial Magnetic Stimulation (HF-rTMS) of the right dorsolateral prefrontal cortex (DLPFC) in treating Post-traumatic Stress Disorder (PTSD) is currently regarded as a level B intervention (probable effectiveness). HF-rTMS has attracted interest as a neuromodulation therapeutic method for PTSD. Prolonged exposure and reactivation therapy are also regarded as first-line treatments for PTSD. Randomized controlled clinical studies examining the effectiveness of several HF-rTMS sessions coupled with psychotherapy have not yet been completed. In total, 102 patients with refractory PTSD will be randomly assigned (1:1) to reactivation therapy, in addition to either active HF-rTMS (20 Hz) or sham HF-rTMS, for 12 sessions in a nationwide, multicenter, double-blind controlled trial. The impact on PTSD symptoms and neurocognitive functioning will be assessed. The primary outcome is the PTSD severity score measured based on the Clinician-Administered PTSD Scale (CAPS-5) at one month. If this additional therapy is successful, it may strengthen the case for regulatory authorities to approve this additional technique of treating PTSD. Additionally, it expands the field of neurostimulation-assisted psychotherapy.

## 1. Background

Post-traumatic stress disorder (PTSD) is defined as a set of symptoms that occur after being exposed to a traumatic incident (re-experiencing the event, avoidance, negative thoughts or sensations, trauma-related arousal, and reactivity). PTSD is ranked as one of the top ten public health issues by the World Health Organization (WHO) [1]. In France in 2014, its frequency was predicted to be 2.4% among civilians [2]. PTSD is frequently a chronic disorder, with just one-third of patients healing after a one-year follow-up and one-third still experiencing symptoms 10 years later [3]. PTSD is associated with significant disability, medical illness, and premature death. The National Comorbidity Survey [3] reports that 16% of PTSD patients have one coexisting psychiatric disorder, 17% have two, and 50% have three or more. Depressive disorders, anxiety disorders, and substance abuse disorders are two to four times more prevalent among individuals with PTSD. Approved therapies are effective in only 41% of patients, even when combined with specific psychotherapy, and the optimal treatment alternatives for non-responding PTSD patients remain unknown. The current mainstay of treatment relies on trauma-focused psychological interventions and psychopharmacological treatments. For most adults newly diagnosed with PTSD, the first-line treatment is trauma-focused psychotherapy that includes exposure, rather than treatment with a serotonergic agent. However, in many cases, a serotonin reuptake inhibitor (SSRI) is a reasonable alternative for patients who prefer medication to psychotherapy or when cognitive-behavioral therapy is unavailable (a frequent situation). Multiple randomized clinical trials [4] have found that patients with PTSD experience reduced symptoms when treated with an SRI compared to a placebo. The majority of PTSD patients receive pharmacological treatment, with antidepressant agents prescribed in 89% of pharmacological treatment cases [5]. Sertraline and paroxetine are the two approved medications [5] and are considered first-line medication treatments for PTSD. However, 41% of subjects fail to respond to antidepressants.

Traumatic reactivation therapy is effective, but remission is achieved in only 40% of subjects [6]. Much of PTSD’s pathophysiology remains unclear. It encompasses a wide range of neurobiological changes, involving disturbances in the hypothalamic–pituitary adrenal axis, hyperactivation of the amygdala complex, and attenuation of some hippocampal and cortical functions (prefrontal cortex) [7]. Abnormalities in these interconnected regions mediate the acquisition of fear responses in PTSD, avoidance of trauma reminders, impaired emotion regulation (manifested as irritability, anger, or reckless behavior), and the persistence of defensive responses once safety has been restored [5]. Functionally, the metabolic activity of the prefrontal–amygdala cortex circuit in PTSD is significantly altered, with increased activation of the amygdala and decreased activation of the prefrontal cortex [7]. An increasing number of studies investigate innovative therapies targeting brain activity to alleviate PTSD symptoms. A treatment augmenting the activity of prefrontal regions could effectively improve executive control of fear responses, thus improving PTSD. Transcranial magnetic stimulation is a noninvasive brain stimulation procedure that can modulate neuronal activity via the administration of magnetic pulses to specific brain areas.

Preliminary studies suggest that transcranial magnetic stimulation of the right dorsolateral prefrontal cortex has a positive effect [5]. Repetitive Transcranial Magnetic Stimulation (rTMS) provides focused, non-invasive stimulation of cortical areas of the central nervous system through an electromagnetic coil placed in contact with the scalp and connected to a magnetic stimulator that discharges short alternating electrical pulses to the central nervous system. The rTMS can be divided into high-frequency (HF) (5–20 Hz) stimulation, which increases cortical excitability, and low-frequency (LF) (≤1 Hz) stimulation, which suppresses excitability. Moreover, repeated sessions at regular intervals help maintain the effect over time [8], and the FDA has approved rTMS for depression and auditory-verbal hallucination treatment.

Although evidence is limited, HF rTMS may improve PTSD symptoms by modulating the prefrontal cortex activity. The mechanisms by which rTMS might exert its effects to mediate symptom improvement in PTSD remain unclear. The most recent meta-analysis [9] on PTSD selected 18 articles for the systematic review, and 11 were suitable for the meta-analysis. The study aimed to determine the most effective stimulation frequency for PTSD symptoms (assessed with CAPS-5). Unilateral high-frequency rTMS stimulation over the DLPFC was mainly performed on the right side.

The combined sample consisted of 377 subjects (217 in active rTMS groups, and 160 in sham-controlled groups). The total number of female participants was 195. The rTMS was an augmentative treatment, i.e., a previously used drug and/or psychological intervention. The authors conclude that additional research is needed, but both HF and LF rTMS can alleviate PTSD symptoms. HF rTMS could improve the primary and related symptoms of PTSD. HF rTMS demonstrated an improvement in the PTSD total score: pooled SMD (pre-post treatment), 3.24; 95% CI 2.24–4.25 (Yan et al.). The dorsolateral prefrontal cortex (DLPFC) and the dorsomedial prefrontal cortex were the most promising sites of stimulation [9]. Four of the studies (all with rTMS HF) reported that the treatment effect of rTMS was preserved for 14 days to 3 months [10,11,12,13]. Focusing on HF studies [10,11,12,13,14], the total number of pulses for the cure ranged from 4000 [11] to 20,160 [12], with an average of 11,520 pulses. We believe that this figure is too low and justifies conducting a study with more pulses per patient.

Furthermore, two trials [12,15] combined rTMS with trauma-focused psychotherapy (the subject recalls their traumatic event) to determine if the combination has an additive effect. Only Isserles et al. (included in the meta-analysis) used recall cues concurrently with stimulation. The primary outcome measure was the CAPS, measured after four treatment weeks. Mean CAPS scores improved from 88 (±5.5) at baseline to 61 (±8.8) in the EXP-STIM group (N = 9), from 86 (±5.4) to 76 (±10.9) in the NOEXP-STIM group (N = 8), and from 86 (±9.2) to 76 (±10.7) in the EXP-SHAM group (N = 9). Significant improvement was demonstrated in the intrusive component of the CAPS scale in patients administered rTMS after exposure to the traumatic event script. The limited number of subjects encouraged the execution of a larger study on a more homogeneous population (resistant PTSD, for example). The study by Kozel et al. is the largest study to date, with 103 veterans randomized to either 9 active (*n* = 54) or sham (*n* = 49) low-frequency (LF) rTMS sessions (1/week) associated with CPT [15]. The rTMS combined with the CPT group showed greater symptom reductions from baseline on CAPS-5 across CPT sessions (t(df ≥ 327) ≤ −2.01 *p* = 0.023, one-tailed). It was a predominantly military population with different traumas from civilian injuries (sexual assault or serious accidents). LF rTMS sessions were performed before psychotherapy.

The rTMS has shown promise as a potential treatment for PTSD, with some studies suggesting that high-frequency stimulation may improve symptoms by modulating prefrontal cortex activity. However, further research is needed to determine the optimal stimulation parameters and patient populations for this intervention. Additionally, more extensive trials exploring the combination of rTMS with trauma-focused psychotherapy may help elucidate whether these treatments have additive effects on PTSD symptom improvement. Considering Yan’s meta-analysis and Kozel’s results, the heterogeneity in rTMS protocols and the limited number of rTMS studies combining cue exposure make it difficult to draw conclusions about the role of rTMS in PTSD. There is a need for intensive clinical trials in this area. We hypothesize greater efficacy of high-frequency (HF) rTMS compared to low-frequency (LF) rTMS [9], as well as improved synergistic action when rTMS and exposure therapy are performed simultaneously, potentially inhibiting the hyperactive amygdala.

The present study protocol aims to elucidate the effect of multiple sessions of HF-rTMS on PTSD symptoms, cognition, and brain functioning. We will conduct a multicenter, double-blind, randomized controlled trial with 102 PTSD patients randomized (1:1) to either 10 sessions of active HF-rTMS + reactivation therapy or 10 sessions of sham HF-rTMS + reactivation therapy. We will investigate the effect of active versus sham HF-rTMS treatment on measures of symptoms and cognition. We expect reduced symptomatology and improved functioning of neurocognitive tasks in the active HF-rTMS-treated group compared to the sham HF-rTMS-treated group. Our protocol proposes a larger number of pulses than the existing literature. We will evaluate the efficacy of combining neuromodulation through rTMS with script-driven imagery in civilian patients who respond insufficiently to first-line treatments with antidepressants.

## 2. Study Aims

We aim to assess the efficacy of high-frequency (HF) rTMS stimulation over the right dorsolateral prefrontal cortex (DLPFC) in patients with PTSD resistant to a first pharmacological line of treatment. The rationale for focusing on this subgroup of patients is twofold: (1) there is a critical need for alternative treatment options for PTSD patients who do not respond to antidepressants, and (2) in these individuals, combining rTMS with reactivation of traumatic memory may be particularly effective by targeting both the prefrontal cortex (activated by rTMS) and the amygdala (reactivated by script-driven imagery). Our study offers another urgently needed treatment option for PTSD patients who are non-responsive to antidepressants. This is the first clinical research study to combine HF rTMS with memory reactivation in a civilian PTSD population resistant to a first-line of a pharmacological agent.

Pairing rTMS with trauma-relevant cues may represent a novel approach to treating PTSD patients. HF-rTMS over the dorsolateral prefrontal cortex (DLPFC) could correct the deficient cortical activation observed in some treatment-resistant PTSD patients. The efficacy of HF-rTMS treatment may be enhanced by combining it with trauma cue exposure (neuro-enhanced psychotherapy), which would engage the prefrontal cortex during sessions. Restoring the activity of prefrontal regions could effectively enhance the executive control of fear responses, thus improving PTSD symptoms. Therefore, this study aims to investigate the efficacy and underlying mechanisms of 10 add-on HF-rTMS sessions as part of a reactivation therapy protocol for PTSD patients resistant to 6 weeks of first-line conventional treatment.

## 3. Primary Research Objective

The primary objective of this study is to evaluate the efficacy of brain modulation by rTMS, combined with simultaneous traumatic memory reactivation, in patients presenting PTSD resistant to one pharmacological agent compared to sham rTMS, with simultaneous traumatic memory reactivation at one month (M1) after the last rTMS session (V13).

## 4. Secondary Research Objective

The secondary objectives of this study are to assess the efficacy of brain modulation via 20 Hz rTMS targeting the right dorsolateral prefrontal cortex, in conjunction with simultaneous traumatic memory reactivation, on the following aspects:Different dimensions of PTSD assessed using CAPS-5 at M3 and M6 post-treatment;The severity of PTSD assessed using PCL at each visit;Anxiety symptoms at V13 (end of treatment), M1, M3, and M6 post-treatment;Depressive symptoms at V13 (end of treatment), M1, M3, and M6 post-treatment;Changes in social cognition at M3 post-treatment;Changes in quality of life at M1, M3, and M6 post-treatment;The safety of the intervention.

## 5. Methods and Design

### 5.1. Study Design

The TRAUMASTIM study is designed as a national multicenter randomized, double-blinded, shamed controlled trial comparing two parallel arms (1:1): rTMS with simultaneous reactivation of traumatic memory in resistant PTSD versus sham rTMS and simultaneous reactivation of traumatic memory.

### 5.2. Ethical Considerations

This study will be approved by the Medical Ethical Committee (IDRCB No 2021-A02532-39-B). Written informed consent will be obtained before screening for inclusion and exclusion criteria takes place.

### 5.3. Inclusion Criteria Patients Aged between 18 and 65 Years

Presenting PTSD according to DSM-5 criteria;Patient with persistent symptoms (PCL > 44) after 6 weeks of treatment with SSRI;Patient with health insurance (AME excepted);Signed written informed consent.

#### Exclusion Criteria History of Epilepsy or Seizure

Cochlear implants;Cardiac pacemaker or intracardiac lines, or metal in the body;Strong dissociative tendencies, evidenced by an average score > 20 on the DES;Lifetime psychotic or bipolar disorder;Antisocial personality or borderline personality;Brain injury defined by medical report;Current substance dependence (including alcohol, excluding tobacco);Acute suicidal ideation;No adequate mastering of the French language or no ability to consent;Pregnancy (confirmed by a urine beta-HCG) or breastfeeding;Absence of birth control;Patient under legal protection measures and/or deprived of freedom;Participation in any other interventional study or in the exclusion period of any other interventional study.

### 5.4. Intervention

#### 5.4.1. rTMS Protocol

The intervention consists of 12 HF-rTMS sessions in the right DLPFC (rDLPFC) on 12 workdays. The HF-rTMS parameters of the active intervention are as follows: active high-frequency 20 Hz, 3 to 4 sessions of 10 min per week, 2000 pulses/session, and a motor threshold of 110%. In our centers, we regularly offer 3 to 4 rTMS sessions per week for patients with depression, which has proven to be both feasible in terms of logistics and well accepted by the patients receiving the treatment. The number of sessions per week will be determined based on the logistical constraints of the investigating centers and the patients being stimulated. The device used will be an electromagnetic stimulator, which is used to artificially provoke a functional activity (GMDN 13762) CE marked, Class IIa.

The rDLPFC will be located at position F4 using the International 10–20 EEG system or a neuronavigation system. During the stimulation, participants will be seated on a comfortable chair with extra neck support. One stimulation session will take approximately 10 min.

The motor threshold will be determined at rest before the first stimulation session, using single-pulse TMS in combination with motor-evoked potentials. The optimal spot on the scalp for stimulation of the right abductor pollicis brevis muscle will be located, and the motor threshold will be established by delivering single stimulations to the motor cortex. The motor threshold, defined as the lowest stimulation intensity producing five motor evoked potentials (MEPs) of at least 50 μV in 5 out of 10 stimulations, will be measured by gradually increasing the stimulation intensity. The intervention will be applied by an rTMS-trained researcher.

Sham rTMS will be delivered, following the same procedure. The trauma memory reactivation procedure will be the same as that in the experimental group. Sham rTMS will be performed with a special probe for sham procedures. The instruction to use sham rTMS will be the same as that for the active device. A sham coil providing the same noise, heat, clicks, and sensations as the active coil will be used without any magnetic pulse.

#### 5.4.2. Reactivation Therapy

The reactivation therapy consists of the development of a traumatic script where participants will describe their traumatic event during a “script” preparation session, which will be used to reactivate the traumatic memory before each rTMS session. The intervention will be performed by a reactivation therapy-trained physician.

### 5.5. Number of Participating Sites

There will be 8 national sites involved, located in hospitals (outpatient or inpatient settings).

### 5.6. Participant Identification

Participants in this clinical investigation will be identified using the following format: site number (3 digits); sequential enrollment number for the site (4 digits); surname initial and first name initial. This unique reference number will be used throughout the duration of the clinical investigation. A randomization number will also be assigned when the participant is randomized, following the RXXX format, in accordance with the sponsor’s procedures.

### 5.7. Randomization

A computer-generated randomization list (allocation ratio: 1:1) will be created by the Unité de Recherche Clinique (URC) of Saint-Antoine Hospital (AP-HP). Randomization will be block-balanced and stratified by site, with block widths undisclosed to investigators. The rTMS operator will perform randomization through a web interface (CleanWeb Software, Telemedicine, Paris, France) during the inclusion visit or before the first rTMS session.

### 5.8. Blinding Methods and Measures to Protect Blinding

The allocated arm (active or sham rTMS) will not be documented in the medical record. Blinding will be maintained for the patients included and for investigators involved in participant inclusion and follow-up.

Two independent staff members will be designated to receive the randomization email and forward it to the rTMS operator. These individuals and the rTMS operator, who will be aware of the sham or active condition, will not be involved in the participant follow-up or evaluation. A sham coil producing the same noise, heat, clicks, and sensations as the active coil but without any magnetic pulses will be used.

### 5.9. Unblinding Procedures, If Applicable

If deemed essential, the investigator can request unblinding for a specific patient by contacting the DRCI and the DRCI project advisor, whose contact information is provided on the protocol cover page. In this trial, no emergency unblinding request is mandatory.

## 6. Procedure and Data Collection

### 6.1. Inclusion Procedure

Patients will be recruited during an appointment at a participating specialized trauma unit (trauma center). Patients may be either hospitalized or treated as outpatients. During the selection visit and prior to any research-related examination, the physician will propose the study to the patient, informing them about the objective, the nature of the constraints, the computerized processing of the data that will be collected during this research, and their rights of access, opposition, and rectification to these data. During this visit, the eligibility criteria will be verified by the investigator or co-investigator. An information form summarizing these points will be provided to the patient. If they are interested, informed consent will be signed, and the participant will be screened for inclusion and exclusion criteria. If a patient meets all the inclusion criteria and none of the exclusion criteria, they will be included in the study by the researcher.

A computer-generated randomization list (1:1) will be created by the Clinical Research Unit (URC) of Saint-Antoine Hospital (AP-HP). Randomization will be block-balanced and stratified on-site. The width of the blocks will not be disclosed to investigators. Randomization will be performed using a web interface (CleanWeb 2.0 Software, Telemedicine, Paris, France) at the inclusion visit or before the first rTMS session by the rTMS operator. The allocated arm (active or sham rTMS) will not be included in the medical records. The blinding of the allocated treatment will be applied to both the included patients and investigators involved in the patient’s inclusion and follow-up. The operator performing the rTMS will be the only one aware of the sham or active condition. This operator will not be involved in the patient’s follow-up and evaluation.

### 6.2. Patient Recruitment

In the event of recruitment challenges, a recruitment assistance service (Cline Research) may be considered to facilitate the dissemination of information about the study to the general public, enable online pre-screening, and potentially involve external referral sources.

### 6.3. Overview of Study

For an overview of the intervention, instruments, order of assessment, and moment of assessment, see Figure 1 and Table 1.

## 7. Outcomes and Instruments

### 7.1. Sample Characteristics

General patient characteristics such as age, handedness, educational level, and use of medication will be assessed. The Mini International Neuropsychiatric Interview (MINI) will be used to determine whether the participant has any psychiatric DSM-IV diagnoses.

### 7.2. Primary Outcome Measure

The primary assessment criterion is the PTSD severity score measured using the Clinician-Administered PTSD Scale (CAPS-5) at the one-month follow-up (V13).

### 7.3. Secondary Outcome Measures

PTSD severity scores at M3 and M6 (repetition, avoidance, and neurovegetative activation) measured using CAPS-5 (structured interview);PTSD severity score measured using the PTSD Checklist (PCL-5 self-questionnaire) at M1, M3, and M6;Dissociative symptom severity scores assessed by the Clinician-Administered Dissociative States Scale (CADSS), the Multidimensional Assessment of Interoceptive Awareness (MAIA), and the Dissociative Experiences Scale (DES) at V1, V13, M1, M3, and M6.The severity scores of the dimensions of anxiety (measured using HAM-A) and depression (measured using HAM-D) at M1, M3, and M6.Social cognition at M0 and M3 assessed using the Eckman test, “reading the Mind in the Eyes” test (Baron-Cohen, test des Faux-Pas (Baron-Cohen), Empathy Quotient questionnaire (Baron-Cohen), and Toronto Alexithymia Scale (TAS-20).Quality of life assessed using WHOQOL EuroQol Health Measure, EQ-5D-5L;Proportion of adverse events during the active treatment phase and during the follow-up.

## 8. Statistical Analyses

### 8.1. Power Analysis

The current study is the first to investigate the effect of multiple reactivation therapies enhanced by 20 Hz HF-rTMS sessions on PTSD score severity six months after the last stimulation session. To the best of our knowledge, no studies have investigated the effect of rTMS on PTSD after such a period. A detailed analysis plan will be defined before the database lock. The analysis will be performed by a statistician from URC-Est after data review and database lock. Therefore, no interim analysis is planned. Analyses will be performed using SAS^®^ software (version 9.4 or the updated version).

Based on the literature, we estimate that the CAPS total severity score at M1 in the sham rTMS + memory reactivation procedure group will be at 50 ± 24 (S5 value from [15]). Assuming a 15-point reduction in the experimental group to indicate a clinically significant difference based on the CAPS manual, two-sided alpha = 5% and beta = 15%, and a standard error of 24 in both groups, 92 patients are needed (Student *t*-test, East 6, Cytel). Considering 10% of non-evaluable patients, 102 randomized patients are needed. We anticipate fewer non-evaluable patients than in Kozel’s study (4). Our reactivation memory is less demanding for patients than that of Kozel’s program. Regarding the CAPS score at inclusion, we anticipate a comparable score for the overall severity. This score is close to that reported in Yan’s meta-analysis (3) on the efficiency of rTMS in PTSD. There is empirical evidence regarding the value of prolonged exposure to a wide range of injuries. However, clinical trials have shown that less than 50% of patients improve with exposure therapy and only 15% achieve complete remission [16]. In our study, we will include a more resistant population that is classically less sensitive to the effects of psychotherapy alone. All tests will be two-sided, and a *p*-value of <0.05 will be considered significant.

The baseline characteristics of the patients will be described overall and for each group. Qualitative data will be described with frequencies and percentages and quantitative data will be described with mean and standard error or with median and interquartile interval according to the distribution of the variable.

### 8.2. Primary Criterion Analysis

An analysis will be conducted based on the intent-to-treat (ITT) population.

The CAPS total severity score at M1 will be compared between groups using the Student *t*-test. In the case of a non-normal distribution, transformation will be performed. Sensitivity analyses on available data and on per protocol population using the same methods will be performed.

### 8.3. Secondary Criteria Analysis

PTSD severity scores at V15 and V16 (repetition, avoidance, and neurovegetative activation) measured with CAPS (structured interview) will be described by mean and standard deviation, and the mean difference between groups will be calculated using its 95% confidence interval (CI). Each score (PCLS self-questionnaire, HAM-A, and HAM-D scores) will be analyzed in the same way at V14, V15, and V16. Graphical representations of the evolution over time will be performed. The mean difference between groups and 95% CI will be calculated.

Based on the PTSD severity score measured using the PTSD Checklist (PCLS self-questionnaire), a linear mixed regression model may be performed with the patient as a random effect and the group as a fixed effect. In the case of a non-normal distribution of the interest variable, a transformation will be performed.

Social cognition at V0 and 15 will be assessed by the Eckman test, “Reading the Mind in the Eyes” test [17], Test of Faux-Pas [18], Empathy Quotient questionnaire [19], and Toronto Alexithymia Scale (TAS-20) [20]. The demonstration of cognitive improvement requires a free interval between the end of the active treatment phase and the completion of the assessment. The difference between the M3 and V0 measurements of each test will be calculated using their 95% CI. The evolution of quality of life (assessed using the WHOQOL [21] and EuroQol Health Measure, EQ-5D-5L [22]) will be analyzed. Each subscore of the two scales will be described and analyzed in the same way.

The proportion of at least one adverse event observed during the active treatment phase and during the follow-up will be described and compared between groups using Fisher’s exact test. The proportion difference between groups and their 95% CI will also be calculated. Each type of adverse event will also be described by the group.

## 9. Discussion

This article presents a multicenter double-blind randomized clinical trial protocol investigating whether 12 sessions of reactivation therapy enhanced by active 20 Hz HF-rTMS, compared with the same reactivation protocol with sham HF-rTMS, improves the treatment outcomes of PTSD. The choice of 12 sessions and 20 Hz was justified by a review of the literature, and in particular by a recent meta-analysis [23], which found several studies to be effective with 12 sessions and also found that the most effective study was Ahmadizadeh’s 20 Hz study [24]. The aim of this study is to decrease symptom severity and improve brain functioning measures, such as social cognition. Participants, who are non-responders to medications, will be offered a promising treatment alternative. The main expected individual benefit would be a regression of core PTSD symptoms: re-experience of the event, avoidance, negative thoughts or feelings, trauma-related arousal, and reactivity. Reducing PTSD symptoms has been largely reported in large cohort studies on morbidity and mortality. An improved quality of life is also expected.

One of the main challenges of this study will be the completion of the entire follow-up procedure since participants will be called at three and six months to assess symptom severity. The strength of this study is that it investigates the effect of multiple HF-rTMS sessions in order to enhance reactivation therapy. So far, only a few studies have looked at neuro-enhanced psychotherapy.

If this study reveals decreased severity scores in the active stimulation group compared with the sham stimulation group, and the active HF-rTMS induces negligible side effects, this may lead to a shift in PTSD treatment and a need for larger clinical trials on neuro-enhanced psychotherapy for other disorders. Furthermore, future studies will be needed to compare the sensitivity and specificity of our protocol against standard treatments. This could eventually result in approval by regulatory authorities as an additional treatment method for PTSD; for instance, the Food and Drug Administration has approved HF-rTMS for the treatment of depression or OCD [25].

## Figures and Tables

**Figure 1 brainsci-13-01274-f001:**
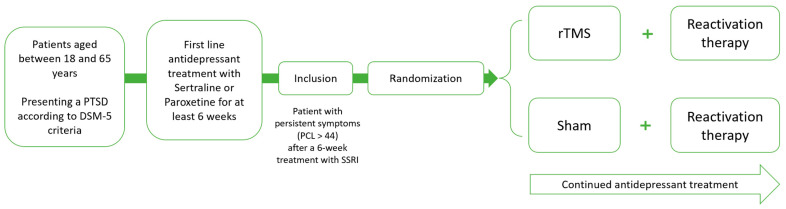
Overview of the intervention.

**Table 1 brainsci-13-01274-t001:** Overview of instruments and assessment.

Visit	0	1	2	3	4	5	6	7	8	9	10	11	12	13	14	**15**	**16**
PDI	✕																
HCG		✕															
Active/Sham rTMS			✕	✕	✕	✕	✕	✕	✕	✕	✕	✕	✕	✕			
Trauma Script		✕															
Reactivation			✕	✕	✕	✕	✕	✕	✕	✕	✕	✕	✕	✕			
MINI-5		✕													✕		
CAPS-5	✕														✕	✕	✕
PCL-5	✕		✕			✕			✕			✕		✕	✕	✕	✕
CGI			✕			✕			✕			✕		✕	✕	✕	✕
Eckman Test		✕														✕	
Empathy Quotient		✕														✕	
Baron-Cohen		✕														✕	
TAS-20		✕														✕	
HAM-D		✕												✕	✕	✕	✕
HAM-A		✕												✕	✕	✕	✕
WHOQOL		✕													✕	✕	✕
EQ-5D-5L		✕													✕	✕	✕

## Data Availability

Data will be available on demand.

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
