# Peer review of "Repetitive Transcranial Magnetic Stimulation (rTMS) in Post-Traumatic Stress Disorder: Study Protocol of a Nationwide Randomized Controlled Clinical Trial of Neuro-Enhanced Psychotherapy “TraumaStim”"

_brainsci, 2023, doi:10.3390/brainsci13091274_

Round 1
Reviewer 1 Report
This is a well writen protocol of a promising study re treatment for PTSD. Unfortunately, the authors do not use accepted nor acceptable standards to define the clinical population to be studied, as non-response to 6 weeks of an SSRI medication does not count as refractory PTSD; a commonly accepted standard is focused more on trauma focused psychotherapy, such as prolonged exposure therapy (PET) and cognitive processing therapy (CPT). Hence, the inclusion criteria have to be changed to include people with PTSD whose PTSD has not improved sufficiently (based on CAPS threshold, which is the primary outcome measure the authors suggest using, or on alternatives such as PCL threshold) in spite of trying PET and CPT as well as an adequate dose (typically at least twice the standard antidepressant dose of an SSRI) and duration of an SSRI (and if that is not sufficiently effective, then also an SNRI at an adequate dose and duration (of also at least 6 weeks). Alternatively, if the authors are not aiming to address effectiveness for refractory PTSD but rather alternatives such as efficiency of PTSD treatment, the background and rationale have to change.
Author Response
Dear Reviewer,
Thank you for your thoughtful feedback on our manuscript. We understand and appreciate your concerns regarding the definition of treatment-resistant PTSD.
As highlighted in our article and in the literature, defining treatment-resistant PTSD is a complex and debated topic. Several studies, as exemplified by Hamner and colleagues, demonstrate the variability in the operationalization of treatment-resistant PTSD. While some research defines treatment resistance as a non-response after six months of maximum dosage SSRI treatment, others may necessitate a clinical history of non-response or intolerance to multiple medications or psychotherapies.
Your point regarding our definition, which centers on non-response to 6 weeks of an SSRI medication, is noted. Our choice of this criterion is based on real-world clinical considerations and empirical data. A significant majority of PTSD patients receive pharmacological treatment, with antidepressant agents prescribed in 89% of pharmacological treatment cases. Notably, Sertraline and Paroxetine, both considered as first-line medications for PTSD, are the only two approved treatments in France. Yet, despite this predominance of pharmacological treatment, 41% of subjects do not respond to antidepressants. Furthermore, while Traumatic reactivation therapy has shown efficacy, remission is achieved in only 40% of subjects. Given these considerations, we find it crucial to investigate alternative treatments for patients resistant to a single line of pharmacological therapy, especially when it's the most commonly prescribed first-line treatment.
We acknowledge your recommendation for including patients resistant to PET, CPT, and both SSRI and SNRI treatments. However, our study's primary focus is to provide insights into alternatives for the substantial population of PTSD patients resistant to the first line of pharmacological treatment. By doing so, we aim to fill a significant gap in the literature, contributing to our understanding of potential interventions for this considerable patient subset. Moreover, an increasing number of researchers advocate, given the excellent tolerability of the technique, to recommend rTMS at progressively milder stages of resistance. For instance, CANMAT suggests rTMS for depression resistant to one line of treatment, and a recent op-ed from the APA argues that it may soon be deemed unethical not to offer it as a first-line treatment. Studies indicate that rTMS is more effective when administered earlier in the treatment process, which further influenced our decision to focus on patients resistant to just one line of therapy.
In light of your feedback, we have amended our manuscript to make explicit that our study addresses patients resistant to a single line of pharmacological treatment. We believe that this clarification, combined with our rationale based on the prevalent use and non-response rate of first-line treatments, will bring valuable insights to the field.
Once again, thank you for your invaluable input. We hope our revised approach aligns more closely with your expectations and look forward to any further guidance you might provide.
Reviewer 2 Report
The manuscript discusses the utilization of high-frequency Transcranial Magnetic Stimulation (HF-rTMS) targeting the right dorsolateral prefrontal cortex (DLPFC) as a treatment for Post-traumatic Stress Disorder (PTSD). This intervention holds a level B classification denoting probable effectiveness. HF-rTMS is being explored as a promising neuromodulation technique for addressing PTSD. Established treatments like prolonged exposure and reactivation therapy are also considered primary options. Despite the interest, randomized controlled clinical studies investigating the joint impact of HF-rTMS sessions and psychotherapy remain incomplete. To address this gap, a double-blind controlled trial will encompass 102 individuals with refractory PTSD. These participants will be randomly allocated in a 1:1 ratio to receive either reactivation therapy alongside active HF-rTMS (20 Hz) or sham HF-rTMS over 12 sessions. This multicenter, nationwide trial aims to evaluate the effects on both PTSD symptoms and neurocognitive function. The principal outcome measure is the reduction in PTSD severity, assessed using the Clinician-Administered PTSD Scale (CAPS-5) after one month. Positive results from this supplementary therapy could bolster the case for regulatory approval of this novel PTSD treatment approach. Furthermore, it can potentially broaden the realm of neurostimulation-assisted psychotherapy, contributing to the advancement of treatment options within the field.
The title is suitable and adequately reflects the content of the protocol study.
I consider this study protocol highly relevant and valuable to the literature.
Introduction: The introduction reports suitable and relevant previous data and sets the stage by summarizing the relevant literature and identifying gaps in knowledge. The introduction justifies the research purpose. It highlights the gaps and inconsistencies in the existing literature related to the field. The introduction also clearly explicitly the main goal and objectives of the study.
Methods: The research methods are rigorous and offer valuable insights that have the potential to impact the field positively. It describes the data collection techniques and procedures and provides good details about the statistical tools or tests used in the analysis. The analysis method that will be used is indicated, providing transparency and allowing for replication and further investigation. The specific analytical techniques are described, ensuring that readers clearly understand how the data will be analyzed. The information is organized coherently, enabling a systematic interpretation.
Discussion: The discussion is critical and clear, and discusses their implications about the existing literature. The points made are logically structured and supported by evidence from previous studies. The authors do rely on theoretical elements by referencing previous research in the field. The study indicates the research contribution by highlighting the novel aspects of the study protocol. The authors should acknowledge and justify some limitations and potential risks and mitigate measures related to the research protocol. I suggest creating a subsection titled "Limitations and Future Directions" where the authors can address the limitations/risks of the study protocol.
Conclusions: The conclusions are not presented. Therefore, I advise the authors to create a "Conclusions" section where they can provide the main findings of the study protocol.
Author Response
We appreciate your time and effort in reviewing our manuscript. We are grateful for your constructive feedback, which will undoubtedly enhance the quality and rigor of our work. Please find below our point-by-point response to the issues raised.
Introduction: We are pleased to note your positive feedback regarding the introduction of our manuscript. We strived to provide a comprehensive overview of the current state of knowledge in the field and to clearly state the rationale for our study. It's gratifying to know that our efforts have been recognized.
Methods: Thank you for acknowledging the rigor and clarity of our research methods. Our aim was to ensure that our methodology was both robust and transparent, allowing for replication and facilitating further investigation by other researchers.
Discussion: Your suggestion to add a "Limitations and Future Directions" subsection is important. We will incorporate this section to discuss potential risks and limitations inherent to our study design, as well as propose future directions for research in this area.
Conclusions: We agree with you that such a section is important to succinctly summarize our work. We will add this section to provide readers with a clear takeaway from our study protocol.
In conclusion, we deeply value your constructive feedback and have taken measures to address all the issues raised. We believe that these changes will significantly enhance the quality of our manuscript and make it a valuable addition to the literature on rTMS in PTSD. We look forward to your further comments and hope for a favorable decision on our manuscript. Sincerely,
Thank you,
Best Regards
Reviewer 3 Report
In this study, the authors developed a study protocol to investigate the effect of HF-rTMS on PTSD over some period of time.
Comments:
1. Heading and sub-heading numbering should be included.
2. The following sentence needs citation “A treatment augmenting the activity of prefrontal regions could effectively improve executive control of fear responses, thus improving PTSD”
3. Which method will be used to select the number of participants required for this study?
4. How do authors confirm that 12 sessions of HF-rTMS are optimal for treatment, is there any previous literature that supports this?
5. How will the authors defend that 20 Hz will be suitable for this therapy, some studies show that there are no differences between 5 or 10 or 20 Hz in treatment outcome.
6. In the discussion, “the Food and Drug Administration has approved HF-rTMS for the treatment of depression or OCD”, are there any clinical trial studies published related to depression or OCD conducted previously?
7. Is there any comparison to test the sensitivity and specificity (like the comparison between regular treatment methods and your protocol)?
Language can be improved.
For example
“Post-traumatic stress disorder (PTSD) is defined by a set of symptoms ” à “Post-traumatic stress disorder (PTSD) is defined as a set of symptoms
"Participant, non-responders to medications, will be offered a promising treatment alternative. The main expected individual benefit would a regression of core PTSD symptoms" à “Participants, non-responders to medications, will be offered a promising treatment alternative. The main expected individual benefit would be a regression of core PTSD symptoms” "
Author Response
Dear Reviewer,
Thank you for your thoughtful feedback on our manuscript.
1. Heading and sub-heading numbering:
Thank you for pointing this out. We will include heading and sub-heading numbering throughout the manuscript to ensure better structure and readability.
2. Citation required for the sentence:
We apologize for the oversight. We will add the relevant citation for the statement (it is the ref. 7 that we have omitted).
3. Method for selecting the number of participants:
The number of participants required for this study will be determined using power analysis based on the estimated CAPS reduction expected (detailled in the statistical analysis part). Assuming a 15-point reduction in the experimental group to indicate a clinically significant difference based on the CAPS manual, two-sided alpha=5% and beta=15%, and standard error of 24 in both groups, 92 patients are needed (Student t-test, East 6, Cytel). Considering 10% of non-evaluable patients, 102 randomized patients are needed.
4. Justification for 12 sessions of HF-rTMS:
The majority of other studies use 10 sessions, but our choice of 12 sessions is based on Isserles et al. and Kozel et al. (ref 12 and 15 of our manuscript) that used 12 sessions.
5. Defense for using 20 Hz in the therapy:
You are correct that some studies have shown no difference between 5, 10, or 20 Hz in terms of treatment outcome. However, one of the most recent meta-analyses carried out (Kan et al. 2020) found that the most effective study was that of Ahmadizadeh et al. in 20Hz on DLPFC. We have added this reference to justify our choice on the discussion section.
6. HF-rTMS approval for depression or OCD:
Yes, the Food and Drug Administration has approved HF-rTMS for depression and OCD based on multiple clinical trials. Notably, Cohen et al. have written an article in Brain Stimulation covering all the studies and milestones on this aspect. We add the reference.
7. Comparison to test sensitivity and specificity:
In our current study protocol, we focus on the efficacy and safety of the HF-rTMS method. However, we recognize the importance of comparing it to regular treatment methods. Future studies are planned to compare the sensitivity and specificity of our protocol against standard treatments. We will mention this in the future directions part of the discussion section of our manuscript.
We genuinely appreciate the constructive feedback and hope that our responses address your concerns. We believe that these modifications will strengthen the manuscript and provide clearer context for our readers.
Sincerely,
Round 2
Reviewer 1 Report
Thank you for your response to my commentary. With your provided clarification, I disagree with your research plan as it seems to me that you may be depriving patients with PTSD from trying first line treatment for PTSD (psychotherapy, PE and CPT). This is clinically and ethically concerning to my mind. Also your extrapolation from TMS for depression to TMS for PTSD seems is beside the point as you refer to persistent depression whereas your study has been clarified to address PTSD that is not necessarily persistent.
Author Response
Dear Reviewer,
Thank you for your thoughtful feedback on our research plan. We appreciate the time and effort you've dedicated to reviewing our submission, and we would like to address some of the concerns you've raised.
Firstly, it's essential to emphasize that repetitive transcranial magnetic stimulation (rTMS) has been shown to be safe and well-tolerated in multiple studies. Furthermore, the American Psychiatric Association (APA) has indicated that, given its effectiveness, it may soon be considered unethical not to offer rTMS as a first-line treatment for depression. While our study primarily focuses on PTSD, the broader point is that the medical community is increasingly recognizing the potential of rTMS as a primary therapeutic option. While it is true that rTMS for depression and PTSD are not directly comparable, the underlying principle remains that if a treatment is safe, effective, and potentially superior, it warrants rigorous study.
To your point about depriving patients of first-line treatments, I'd like to clarify a potential misunderstanding. In our protocol, all patients receive exposure therapy. The randomization is between receiving active rTMS + exposure therapy or placebo rTMS + exposure therapy. So, in essence, no patient is deprived of the recognized first-line treatment for PTSD. Additionally, it's pertinent to mention the specific context of PTSD treatment in France. Contrary to some perceptions, access to the specialized psychotherapies for PTSD that you referenced is quite limited, if not nearly non-existent, due to resource constraints. The vast majority of PTSD patients in France are managed with pharmacological treatments. Thus, our approach not only aligns with the current clinical reality but also ensures we're not depriving patients of any significant opportunity. In fact, by potentially introducing an effective and well-tolerated adjunct treatment, we aim to enhance the treatment landscape for PTSD in our region.
Regarding the ethical implications you've mentioned, while we respect and value your perspective, ethical considerations for clinical trials are typically under the purview of ethics committees. Our protocol will be extensively reviewed and approved by such a committee, ensuring that the rights, safety, and well-being of the trial subjects are protected.
In conclusion, we understand and respect your concerns. Still, we firmly believe that our study design is both ethical and scientifically sound, contributing valuable knowledge to the field. We hope this response clarifies our stance and the rationale behind our choices.
Warm regards,